# EBV[+] tumors exploit tumor cell-intrinsic and -extrinsic mechanisms to produce regulatory T cell-recruiting chemokines CCL17 and CCL22

Aparna Jorapur[1¤a], Lisa A. Marshall[2¤b], Scott Jacobson[1], Mengshu Xu[3], Sachie Marubayashi[1¤c], Mikhail Zibinsky[4], Dennis X. Hu[4¤d], Omar Robles[4], Jeffrey J. Jackson[4¤e], Valentin Baloche[5], Pierre Busson[5], David Wustrow[4], Dirk G. Brockstedt[1], Oezcan Talay[1], Paul D. Kassner[2]*, Gene Cutler[3¤f]

**1** Discovery Biology, RAPT Therapeutics, Inc., South San Francisco, California, United States of America, **2** Quantitative Biology, RAPT Therapeutics, Inc., South San Francisco, California, United States of America, **3** Computational Biology, RAPT Therapeutics, Inc., South San Francisco, California, United States of America, **4** Drug Discovery, RAPT Therapeutics, Inc., South San Francisco, California, United States of America, **5** CNRS-UMR 8126, Gustave Roussy and Paris-Sud/Paris-Saclay University, Villejuif, France

¤a Current address: Nkarta Therapeutics, South San Francisco, California, United States of America
¤b Current address: Soteria Biotherapeutics, Brisbane, California, United States of America
¤c Current address: Arcus Biosciences, Hayward, California, United States of America
¤d Current address: Genentech Inc, South San Francisco, California, United States of America
¤e Current address: Amgen Inc, Thousand Oaks, California, United States of America
¤f Current address: Ridgeline Computational Biology, San Francisco, California, United States of America
* pkassner@rapt.com

**Data Availability Statement:** All raw RNA-Seq data has been submitted to the NCBI SRA database (accession PRJNA736082). All other relevant data

## Abstract

The Epstein-Barr Virus (EBV) is involved in the etiology of multiple hematologic and epithelial human cancers. EBV[+] tumors employ multiple immune escape mechanisms, including the recruitment of immunosuppressive regulatory T cells (T$_{reg}$). Here, we show some EBV[+] tumor cells express high levels of the chemokines CCL17 and CCL22 both *in vitro* and *in vivo* and that this expression mirrors the expression levels of expression of the EBV LMP1 gene *in vitro*. Patient samples from lymphoblastic (Hodgkin lymphoma) and epithelial (nasopharyngeal carcinoma; NPC) EBV[+] tumors revealed CCL17 and CCL22 expression of both tumor cell-intrinsic and -extrinsic origin, depending on tumor type. NPCs grown as mouse xenografts likewise showed both mechanisms of chemokine production. Single cell RNA-sequencing revealed *in vivo* tumor cell-intrinsic CCL17 and CCL22 expression combined with expression from infiltrating classical resident and migratory dendritic cells in a CT26 colon cancer mouse tumor engineered to express LMP1. These data suggest that EBV-driven tumors employ dual mechanisms for CCL17 and CCL22 production. Importantly, both *in vitro* and *in vivo* T$_{reg}$ migration was effectively blocked by a novel, small molecule antagonist of CCR4, CCR4-351. Antagonism of the CCR4 receptor may thus be an effective means of activating the immune response against a wide spectrum of EBV[+] tumors.

are within the manuscript and its Supporting Information files.

**Funding:** RAPT Therapeutics supported the research conducted by all authors except VB and PB. Employees of RAPT Therapeutics were involved in the design, data collection and analysis, decision to publish, and preparation of the manuscript. All coauthors identified as RAPT Therapeutics employees received a salary from RAPT Therapeutics, including AJ, LAM, SJ, MX, SM, MZ, DXH, OR, JJJ, DW, DGB, OT, PDK, and GC. PB was the recipient of a grant from the Bristol-Myers-Squibb Foundation for Research in Immuno-Oncology n° 1709-04-040.

**Competing interests:** I have read the journal's policy and the authors of this manuscript have the following competing interests: The authors who are identified as current or former employees of RAPT Therapeutics have a potential financial interest in the development of RAPT Therapeutics' FLX475, a drug related to the molecule discussed in this manuscript. No competing interests are reported for the other authors.

## Author summary

The Epstein-Barr Virus (EBV) is associated with many cancers worldwide, including both lymphomas and solid tumors. EBV+ tumors have been reported to have increased numbers of infiltrating regulatory T cells ($T_{reg}$), a cell type that counteracts the body's natural antitumor response. Here we show that EBV+ tumors actually have amongst the highest levels of $T_{reg}$ of all human tumors, as well as having very high levels of the chemokines CCL17 and CCL22, signaling molecules that promote the migration of $T_{reg}$. We found that CCL17 and CCL22 production in different EBV+ tumor cell lines mirrored the levels of production of the EBV protein LMP1, and that the LMP1 gene on its own was sufficient to trigger chemokine expression and $T_{reg}$ migration into a mouse tumor model. Depending on the particular EBV+ tumor type, this CCL17 and CCL22 expression could be coming from the tumor cells themselves, infiltrating host immune cells, or a combination of the two. A recently developed drug that blocks the activity of CCL17 and CCL22 blocked $T_{reg}$ migration into EBV+ and LMP1+ tumors, suggesting that this may be part of an effective treatment for EBV+ tumors in the clinic, helping to reduce the over 140,000 annual deaths from this group of cancers.

## Introduction

The Epstein-Barr Virus (EBV) is one of the most ubiquitous known human viruses, with most individuals infected during childhood or adolescence [1,2]. EBV was also the first virus recognized as oncogenic in humans with the discovery of its role in the etiology of nearly 100% of endemic Burkitt's lymphoma (BL). Since that discovery, EBV has been identified as an important etiological factor in other B-cell lymphomas as well as T and natural killer lymphomas, and epithelial carcinomas including nearly 100% of nasopharyngeal (NPC) and approximately 10% of gastric carcinomas [3–5]. As of 2010, EBV+ tumors were estimated to account for over 140,000 annual deaths globally, with particular impacts in Asia and Africa[6].

In cells latently infected with EBV, the viral genome has the coding potential for 70–80 genes, presenting the potential for numerous foreign antigens to be recognized by the immune system [7,8]. The fact that EBV+ tumors are able to develop suggests that these tumors must have mechanisms for immune escape [8–12]. One such mechanism is the recruitment of regulatory T cells ($T_{reg}$) into the tumor microenvironment (TME). $T_{reg}$ are a subtype of CD4+ lymphocytes that suppress the activity of cytotoxic CD8+ T cells and dampen antitumor immune responses [13]. High $T_{reg}$ infiltrates in various EBV+ tumors have been noted [14–17]. $T_{reg}$ have been shown to be recruited to the TME via C-C motif chemokine ligand 17 (CCL17) and C-C motif chemokine ligand 22 (CCL22; herein described together as CCL17/22) that are expressed directly by some lymphomas or by infiltrating immune cells within the TME [10,18–21]. These chemokines are recognized by $T_{reg}$-expressed C-C chemokine receptor type 4 (CCR4). In fact, a link between EBV infection and upregulation of CCL17/22 expression in lymphomas has been observed and mechanistically linked to the action of the viral latent membrane protein 1 (LMP1) [14,18,19,21]. LMP1 contributes to the transformation and survival of B cells by multiple pathways, including the activation of NFκB—putatively the mechanism for CCL17/22 expression in these cells [20,22]. While expression of these chemokines by immune cells is widely described, the mechanism for CCL17/22 expression in EBV+ tumors of epithelial origin, such as NPC, is less clear.

Here, we provide further evidence for a link between LMP1 expression and CCL17/22 production in EBV+ tumors and demonstrate that this production supports the migration of $T_{reg}$

cells *in vitro* and *in vivo*. That this migration was largely blocked by a novel small-molecule antagonist of the CCR4 receptor, CCR4-351 [23], highlights the importance of the CCL17/22/ CCR4 chemotactic axis in promoting $T_{reg}$ accumulation in these tumors. While RNA *in situ* hybridization (ISH) showed a strong link between EBV-positivity and chemokine expression in Hodgkin lymphoma, a mix of tumor cell-intrinsic and -extrinsic expression of these chemo-kines was revealed in NPC. Human NPC xenografts grown in immuno-deficient mice further demonstrated this mixed tumor-intrinsic and -extrinsic expression of CCR4 ligands. Finally, by evaluating a mouse colon tumor cell line, CT26, engineered to overexpress LMP1, we detected a marked increase in CCL17/22 production by both tumor and dendritic cells accom-panied by an influx of $T_{reg}$. Thus, we have observed varied but convergent mechanisms for CCL17/22 expression that could lead to a TME rich in $T_{reg}$. Treatments that decrease $T_{reg}$ infil-tration or activity, such as a CCR4 antagonist, may be effective immunotherapeutics against multiple EBV+ tumor types.

## Results

To explore the connection between EBV biology and the presence of $T_{reg}$ in EBV+ tumors, we assayed the expression of LMP1, a viral protein reported to be involved in the upregulation of chemokine expression [19,20], and the related LMP2a by Western blot in 9 Burkitt's lym-phoma- or Gastric carcinoma-derived cell lines. Of these, all but KATO III, Ramos, and NCI-N87 have been reported to be EBV+. ELISAs performed on the supernatants of these cell line cultures for the human chemokine proteins, CCL17, CCL20, and CCL22 revealed a very close match between LMP1 expression—observed in Jijoye, Raji, and NC-37 cell lines—and both CCL17 and CCL22 expression by these cells (Figs 1A and S1A). Pearson correlations of 0.96 (p value = 2.3e-6) and 0.95 (p value = 3.4e-5) were found between CCL17 and LMP1, and CCL22 and LMP1, respectively. No correlations with LMP2A were significant (p values > 0.1). CCL20 expression, which has been reported to play a role in EBV-recruitment of $T_{reg}$ [17], was not detected in any sample and is omitted from the figure. LMP2A was detected in all cell lines previously reported as EBV+. NCI-N87, a gastric carcinoma tumor not previously reported as EBV+, also showed expression. This was further confirmed by PCR (S1B Fig), indicating that it actually is EBV+. Unlike LMP1, LMP2A levels did not mirror those of CCL17/22. CCL22 pro-tein levels were proportional to the density of Raji cells in culture (S1C Fig). These results are consistent with prior observations suggesting a contributing role of LMP1 in the expression of CCL17/22 in cells latently infected by EBV, particularly in B cells.

   To further dissect the role of viral proteins in CCL17/22 expression, we transfected Raji cells with LMP1-targeting siRNA (siLMP1), LMP2A-targeting siRNA (siLMP2A), or negative control siRNA. LMP1 protein levels were greatly reduced in the siLMP1-transfected and siLMP1/siLMP2A-cotransfected cells, with no effect of siLMP2A on its own (Fig 1B). Note that an LMP1 reduction by the LMP2A control siRNA was observed–these genes have overlap-ping transcripts that may explain this effect. LMP2A levels were greatly reduced by siLMP2A or siLMP1/siLMP2A, but not by siLMP1 siRNA. Levels of both CCL17 and CCL22 were signif-icantly decreased by either siLMP1 or siLMP2, with stronger reduction in the siLMP1/ siLMP2A combination (Fig 1C). With the previous results, this suggests that while LMP2A expression is not sufficient for CCL17/22 expression in EBV-infected B cells, it does play a role along with LMP1. Interestingly, it has previously been shown that LMP2A cooperates with LMP1 for its activity [24–26].

   In order to demonstrate that the CCL17/22 protein produced by Raji cells is functional, we measured *in vitro* chemotaxis. Control recombinant human CCL22 or supernatant from Daudi or Raji cultures was added to the bottom chambers of migration plates. CCRF-CEM

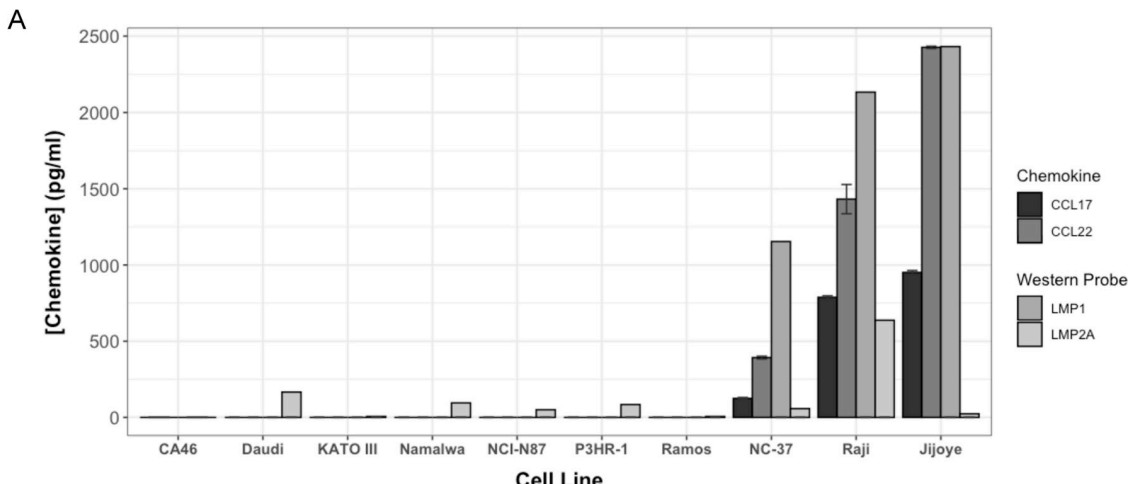

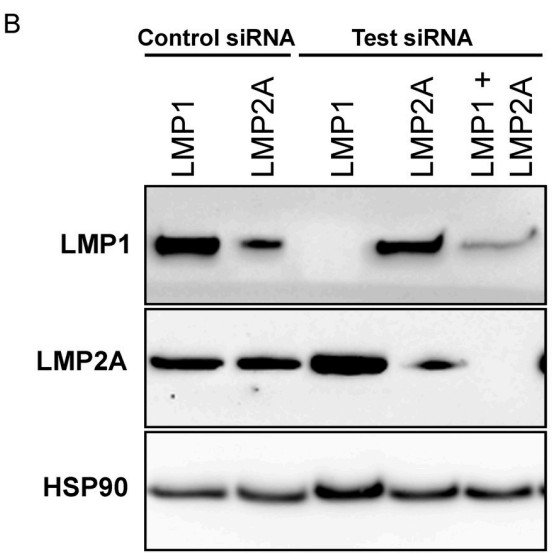

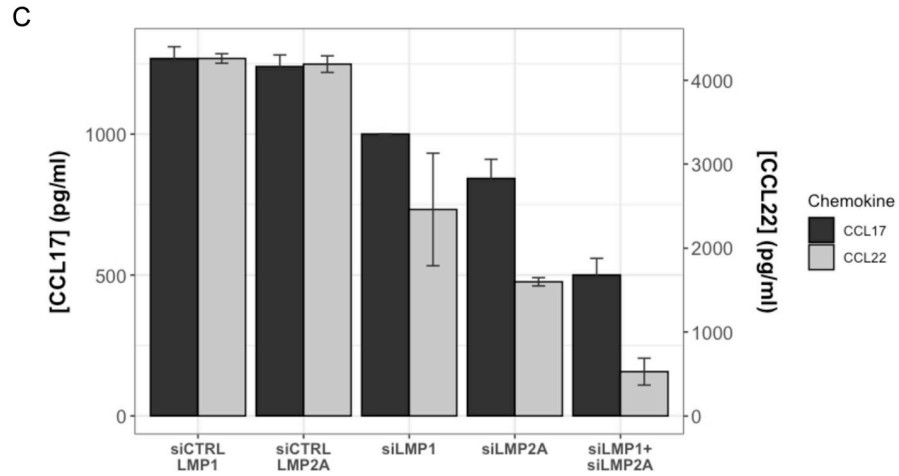

**Fig 1. LMP1 expression is associated with CCL17 and CCL22 production.** (A) Western blots on 50 μg of protein lysate from 9 human cell lines were probed for LMP1 and LMP2A. Supernatants from these cell lines were assayed by ELISA for CCL17 and CCL22 protein. Chemokines were measured in biological duplicates with error bars at ± 1SD. Westerns were quantitated by densitometry and shown at an arbitrary scale with LMP2A protein multiplied by 500 relative to LMP1 for visibility. Cell lines are ordered by CCL22 expression. (B) Western blots on 50 μg of protein lysate from Raji cells 72 hours post transfection with test or control LMP1, LMP2A, or LMP1+LMP2A siRNA were probed for LMP1 and LMP2A protein production, or for respective loading controls, HSP90 and Actin. (C) CCL17 and CCL22 levels in the supernatants of siRNA-transfected Raji cell lines were measured by ELISA.

CD4⁺ T-lymphoblast cells, which express high levels of CCR4, were added to the top chambers followed by quantitation of cell numbers in the bottom chambers after 1 hour. Both recombinant CCL22 and Raji supernatant induced migration of large numbers of CCRF-CEM cells (Fig 2A). This migration was mostly CCR4-dependent, as addition of CCR4-351, a highly specific CCR4 inhibitor, blocked most migration to both recombinant CCL22 and the Raji supernatant. As a more physiologically-relevant assay, the chemotaxis assay was repeated using *in vitro* polarized human CCR4 CD4⁺ cells biased to a T$_{reg}$ phenotype (iT$_{reg}$). A very similar pattern of CCL22- or Raji supernatant-dependent chemotaxis was observed with iT$_{reg}$ (Fig 2B). This chemotaxis could, again, be inhibited by CCR4-351, although the extent of inhibition of chemotaxis to Raji supernatant was less complete.

We sought to further validate this migratory connection between T$_{reg}$ and CCL17/22-producing EBV⁺ Raji in an *in vivo* migration model. We injected Raji or Daudi cells into NOD-SCID mice to generate tumors that grew comparably over the course of 30 days (S2 Fig). ELISA on 19 day tumors recapitulated these *in vitro* chemokine expression patterns: strong CCL17/22 expression was detected in the Raji tumors while Daudi tumors were mostly negative (Fig 2C). We transferred human iT$_{reg}$ into a parallel set of mice 20 days post-tumor cell injection. Seven days post-iT$_{reg}$ transfer, we harvested tumors, and quantitated intra-tumoral iT$_{reg}$ frequency. Since tumor and iT$_{reg}$ both expressed human CD45, migrated iT$_{reg}$ were scored as the fraction of hCD45⁺ cells that were hCD4⁺ hCD19⁻. Migrated iT$_{reg}$ were approximately 3% of hCD45⁺ cells in Raji tumors but were absent in Daudi tumors (Fig 2D). Daily treatment with 50 mg/kg CCR4-351 for 7 days starting 3 hours before iT$_{reg}$ transfer resulted in an 81% decrease of migrated iT$_{reg}$, again demonstrating the key role of the CCL17/22/CCR4 axis in this biology.

EBV-expressed LMP1 mimicry of B-cell CD40 activity is a putative mechanism for upregulation of CCL17/22 in these cells [27,28]. However, upregulation of CCL17/22 is not a reported physiological process in epithelial cells. However, increased T$_{reg}$ have been reported in tumors such as EBV-associated gastric carcinoma (GC) [14] and nasopharyngeal carcinoma (NPC) [16,29]. To put this T$_{reg}$ increase into context, two published NPC RNA-Seq expression data sets were combined with data on thousands of samples from 32 solid tumor types from the Tumor Cancer Genome Atlas (TCGA) and the Therapeutically Applicable Research to Generate Effective Treatments (TARGET) databases. We further subset EBV⁺ GC (GC_EBV) from the TCGA GC samples based on prior annotation [30]. We grouped these samples by tumor type and examined FOXP3, CCL17, and CCL22 expression levels. NPCs and EBV⁺ GC had the first and second highest median expression of FOXP3, a T$_{reg}$ marker, as well as elevated CCL17 and CCL22 compared to the other tumors (Fig 3). Examination of control genes and sample clustering confirmed that the NPC and EBV⁺ GC samples were broadly similar to other tumors (S3 and S4 Figs). The correlation between CCL17+CCL22 and FOXP3 expression within NPC and EBV⁺ GC tumors (Pearson's r = 0.59 and 0.55, respectively) was similar to the correlation across all the TCGA/TARGET samples (r = 0.59).

This high CCL17/22 and FOXP3 mRNA expression in EBV⁺ epithelial tumors is reminiscent of that in EBV⁺ lymphomas [19,21], but this could be due to different underlying

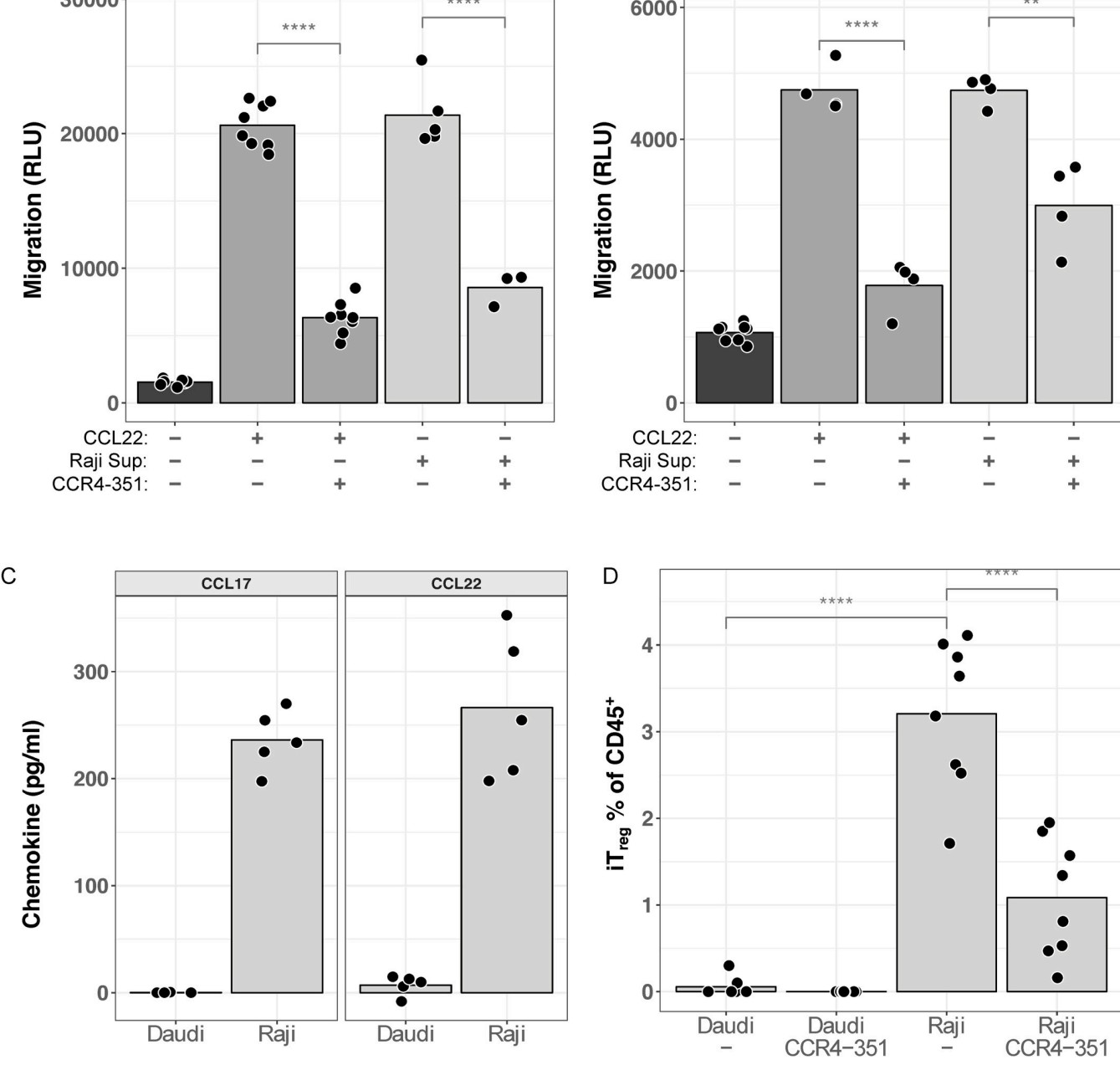

**Fig 2. CCRF-CEM and T_reg migrate towards Raji in a CCR4-dependent manner.** (A) Chemotaxis assays were performed on CCRF-CEM cells towards recombinant CCL22 or Raji cell supernatant with or without the CCR4 antagonist CCR4-351. Relative luciferase units (RLU) are shown as a proxy for migrated cell number. (B) Migration assays were performed with human iT_reg. (C) Raji and Daudi xenograft tumors grown in NOD/SCID mice (n = 5) were assayed for chemokine levels by ELISA. (D) Human iT_reg were transferred into Daudi or Raji tumor-bearing mice (n = 8) and allowed to migrate for 5 days, followed by harvesting of the tumors. Mice were treated with vehicle or CCR4-351. Fractions of human CD45⁺ cells which were iT_reg (identified as CD45⁺ CD4⁺ CD19⁻) were quantitated by FACS.

mechanisms. We profiled 52 Hodgkin lymphoma (HL) biopsies and 15 NPC biopsies by RNA *in situ* hybridization on the RNAscope platform [31], probing for EBER1, a constitutively expressed EBV transcript, along with CCL17, CCL22, and FOXP3. Twenty-seven percent of the HL samples, representing 15 tumors, were EBV⁺ by EBER1 staining (representative

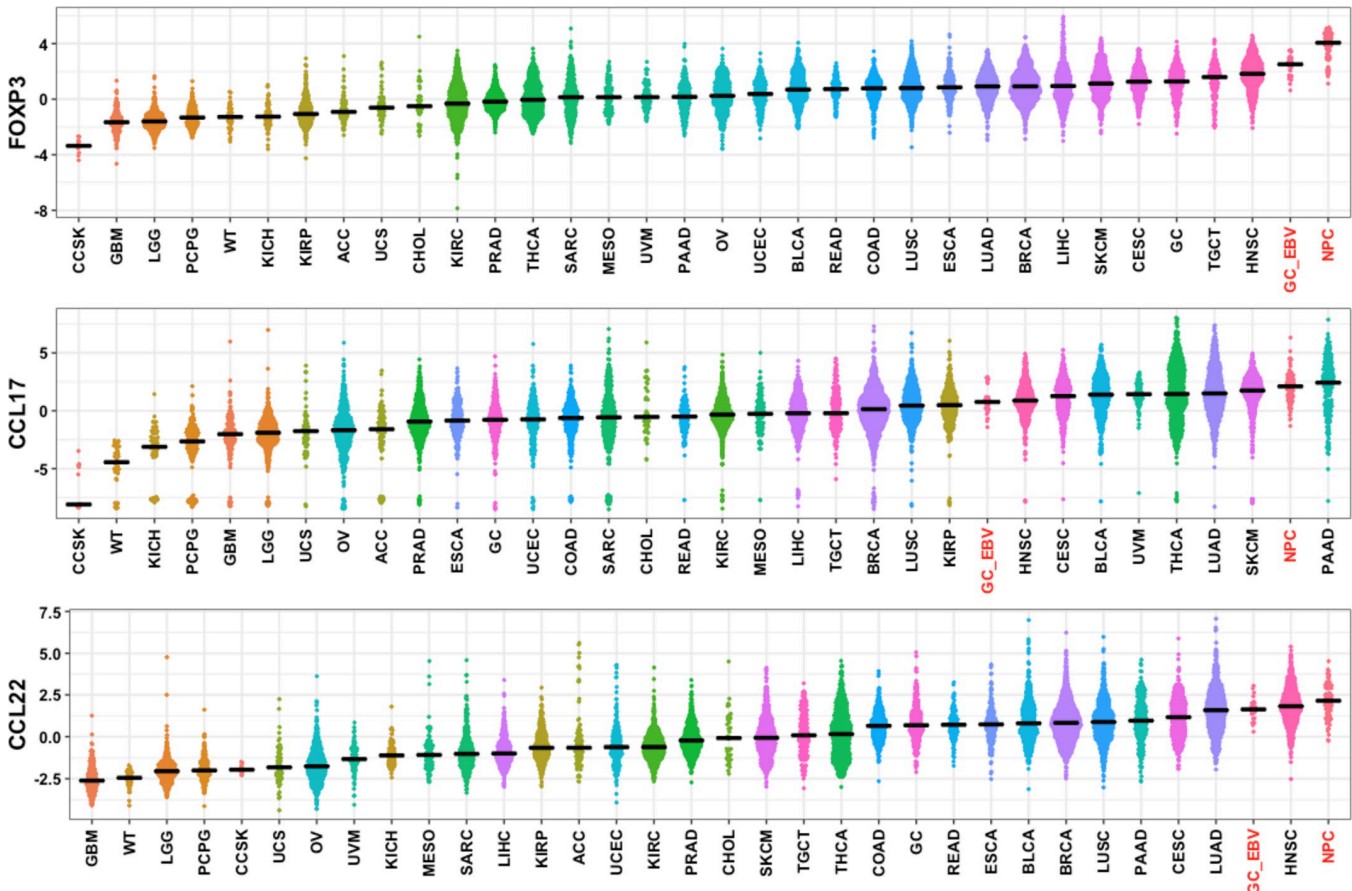

**Fig 3. FOXP3 and CCR4 ligand expression is elevated in EBV⁺ tumors.** (A) RNA-Seq data from 76 Nasopharyngeal carcinomas (NPC) was normalized with data from TCGA and TARGET and analyzed for expression of genes of interest. TCGA Gastric carcinoma samples were divided into EBV⁻ (GC) and EBV⁺ (GC_EBV) subsets. EBV⁺ tumor types NPC and GC_EBV are highlighted in red. Other tumor abbreviations are defined in S1 Table. Tumor types are sorted by increasing median expression and plotted as log₂ Transcripts per Million.

positive staining is shown in Figs 4A and S5). These showed chemokine expression coincident with EBER1 in the HL Reed-Sternberg cells, with 22 out of 26 cores having statistically-significant coexpression of EBER1 with CCL17 and 21 cores with coexpression of EBER1 with CCL22 (FDR < 0.05 by chi-square). The NPC samples also showed chemokine expression associated with EBV-positivity, but quantitation was not possible due to the intensity of the EBER1 staining (representative image in Fig 4B). NPC chemokine expression was not exclusive to EBER1⁺ cells and it was difficult to determine the fraction of CCL17/22 expression that was tumor-extrinsic. CCL22 expression was more clearly observed in the FOXP3/CCL22-stained NPC sections (Fig 4C). FOXP3⁺ cells were seen in the vicinity of CCL22-positivity, which had a punctate pattern throughout the tumor. To discriminate between chemokine expression sources, human NPC tumors grown as mouse xenografts were probed for EBER1 and human or mouse CCL22 transcripts. Of the four xenografts analyzed, C15, C17, C18, and C666-1, only C15 expresses LMP1 [32–35]. Most cells in the C15 (Fig 5A), C17 (S6A Fig), C18 (S6B Fig), and C666-1 (Fig 5B) xenografts stained positive for EBER1. Varying levels of hCCL22 expression throughout the xenografts were observed in C15, C17, and C18, while C666-1 showed mostly punctate expression along with some very-strongly CCL22-expressing cells (Fig 5B). Mouse CCL22 was observed in all samples, but rarer in C666-1, and mostly

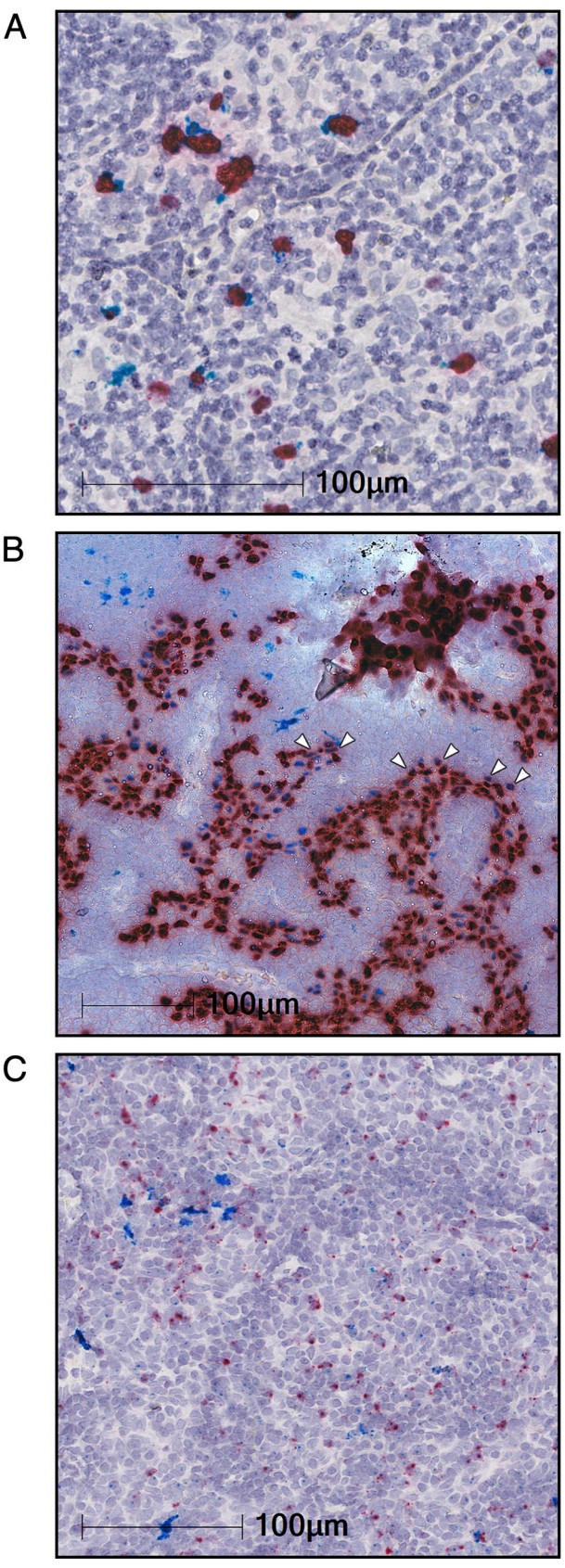

**Fig 4. RNA ISH shows CCL22 and FOXP3 associated with EBV infection in Hodgkin lymphoma and Nasopharyngeal carcinoma.** Representative images of RNA *in situ* hybridization (ISH) of (A) Hodgkin lymphoma (HL) and (B) nasopharyngeal carcinoma (NPC) samples probed for EBER1 (red) and CCL22 (cyan) are shown. Arrowheads in (B) highlight selected EBER1/CCL22 double-positive cells. (C) A matched NPC section serial to that in (B) was probed for FOXP3 (red) and CCL22 (cyan). Nuclear haematoxylin staining is shown in pale blue in all slides. Staining has been digitally enhanced for clarity (unenhanced images in S7 Fig).

confined to regions of EBV⁻ cells (Fig 5A and 5B, right panels). These data demonstrate a mixed pattern of tumor-intrinsic and -extrinsic CCL17/22 expression in NPCs.

To further dissect EBV⁺ epithelial tumor biology, we engineered the mouse colon tumor line CT26 to express LMP1 (CT26-LMP1) as well as a control for the increased antigenicity of LMP1, CT26 expressing chicken ovalbumin (CT26-OVA). None of the cell lines produced mouse CCL17 or CCL22 protein *in vitro* (CCL22 shown in Fig 6A). Tumors formed by CT26 cells contained a modest amount of CCL22, which increased in CT26-OVA tumors, and increased further in CT26-LMP1 tumors (Fig 6A). GFP-marked mouse iT$_{reg}$ were transferred into these tumor-bearing mice for *in vivo* migration. Seven days post-transfer, virtually no iT$_{reg}$ were observed in CT26 tumors and a modest number in CT26-OVA tumors, while a marked increase in iT$_{reg}$ infiltration occurred in CT26-LMP1 (Fig 6B). This was completely abrogated by dosing mice with the CCR4 antagonist, CCR4-351. No changes in mouse iT$_{reg}$ migration to spleen were observed in any of these conditions (S9 Fig).

CT26, CT26-OVA, and CT26-LMP1 tumors were subjected to single cell RNA-sequencing and epitope labeling using the 10x Genomics RNA and CITE-Seq platform [36]. Four major classes of cell types were observed in these tumors–tumor, stroma, lymphoid, and myeloid cells–based on examination of cluster-specific genes, and visualization by uniform manifold approximation and projection (UMAP; Figs 7A, S10–S12, S2 and S3 Tables). CCL17/22 expression was observed in only tumor and myeloid cells (Figs 7A, S10 and S12). Myeloid expression was primarily restricted to classical tissue-resident dendritic cells (cDC) and migratory dendritic cells (mDC) as defined by Binnewiess *et al* [37] and Miller *et al* [38] (Fig 7A). These mDCs also matched the expression pattern described for "mature DCs enriched in immunoregulatory molecules" (mregDCs) [39] (S13 Fig, S4 Table). CCL17/22 expression was also observed in a small number of macrophages (S14 Fig). Tumor cells did not separate into clear clusters and CCL17/22 expression was not biased to any particular region of the tumor cell UMAP (S12 Fig). These four CCL17/22-expressing cell types collectively comprised 92% (CT26), 88% (CT26-OVA), and 86% (CT26-LMP1) of all cells sampled (Fig 7B). However, when counting only CCL17/22-expressing cells, as defined by Transcripts per Million > 1.0, only 0.19% (CT26), 0.86% (CT26-OVA), and 1.3% (CT26-LMP1) were chemokine positive (Fig 7C). The mDC population, which was only 0.026% of all cells in the CT26-LMP1 tumors, accounted for 31% of chemokine-expressing cells. Due to differing expression levels of CCL17/22 across cells (S14 Fig), summing the chemokine expression revealed a 4.1x increase in CT26-OVA and a 10x increase in CT26-LMP1 net chemokine RNA (Fig 7D). In CT26-LMP1, the small number of mDC cells was responsible for the majority of chemokine expression. This combination of CT26-LMP1 tumor cell-intrinsic expression and tumor cell-extrinsic expression from infiltrating myeloid cells, particularly mDCs, is reminiscent of the mixed expression in the NPC xenografts.

## Discussion

The treatment of tumors with immune-based therapies, known as immuno-oncology (IO), holds great promise. There are a multitude of approaches from protein therapeutics such as the approved anti-PD-1, anti-PD-L1, and anti-CTLA-4 checkpoint antibodies to cell-based

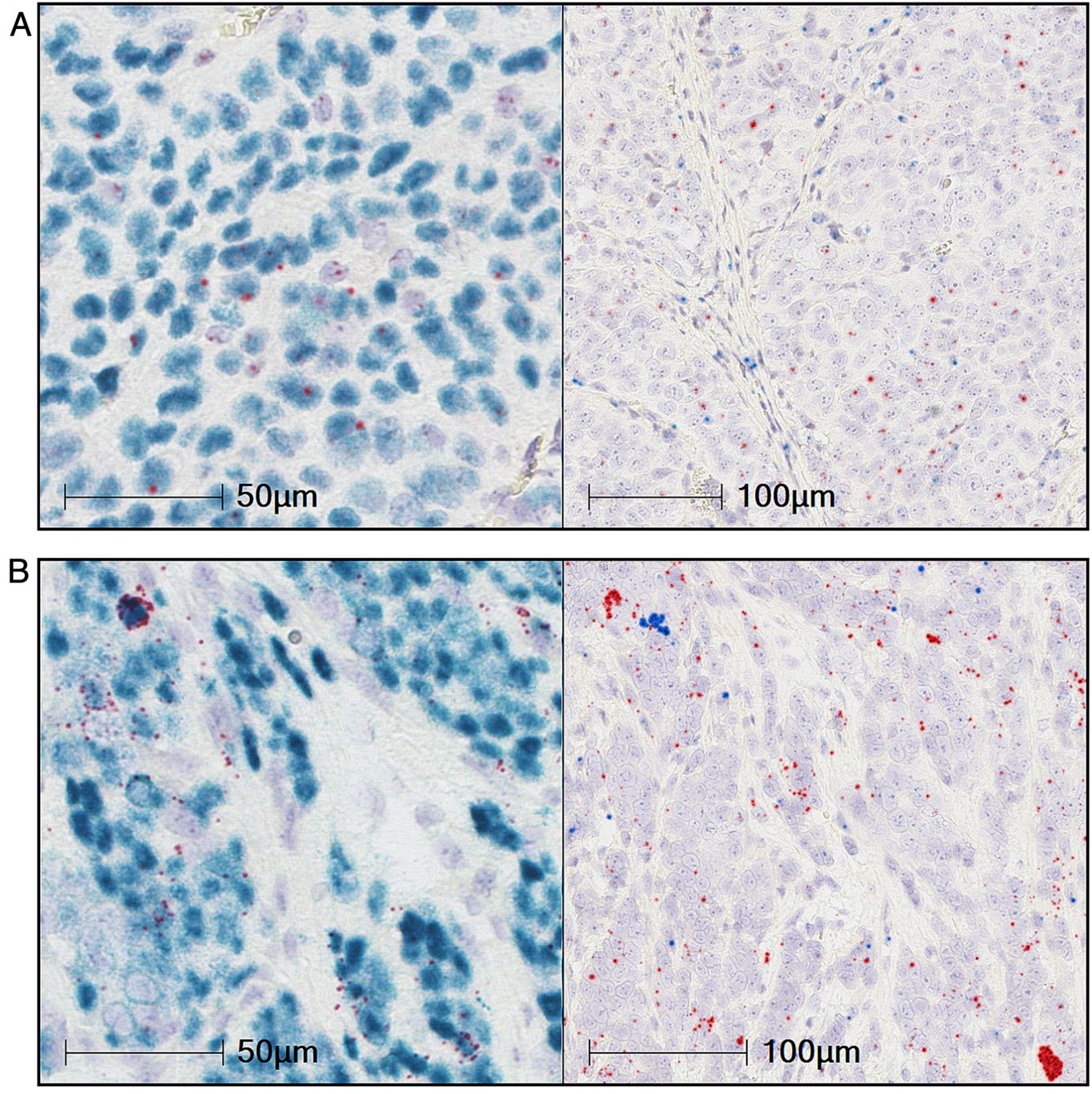

**Fig 5. RNA ISH shows tumor-intrinsic and -extrinsic expression of CCL22 in Nasopharyngeal carcinoma xenografts.** Representative images of RNA *in situ* hybridization (ISH) of a (A) C15 NPC xenograft and a (B) C666-1 NPC xenograft are shown. Left panels are EBER1 (cyan) and human CCL22 (red); right panels are mouse CCL22 (blue) and human CCL22 (red). Nuclear haematoxylin staining is shown in pale blue. Staining has been digitally enhanced for clarity (unenhanced images in S8A and S8B Fig).

therapies and small molecule therapies. These therapies are each expected to be active only against tumors with particular immunologic or antigenic phenotypes. We would expect that EBV-driven tumors, which should be particularly immunogenic due to the expression of foreign viral antigens, to employ particular immune-evasive strategies that could be targeted by

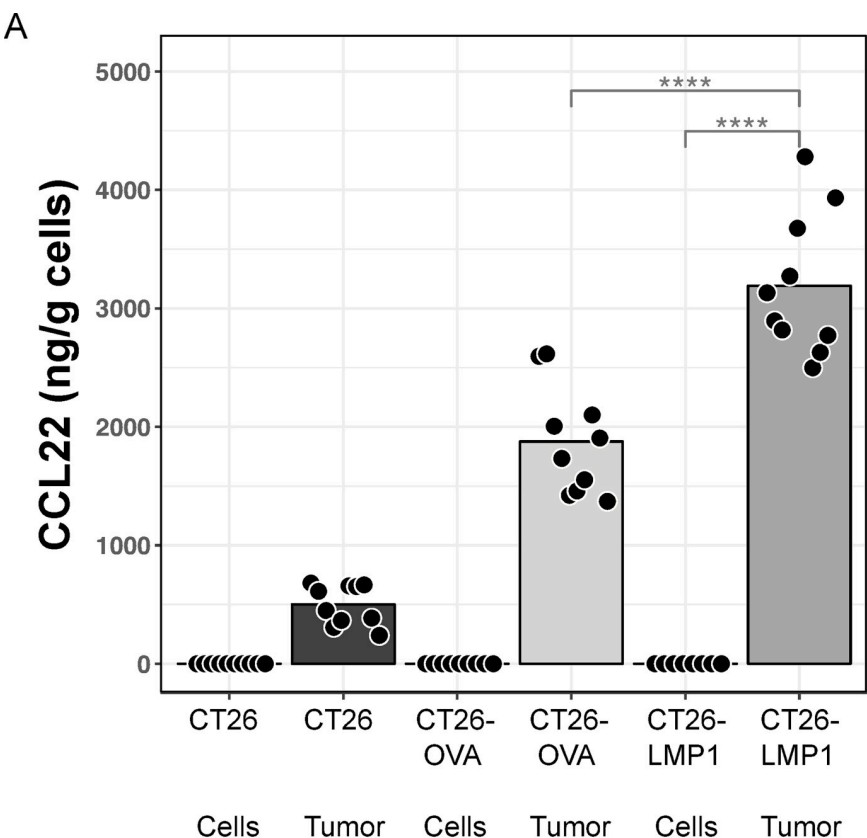

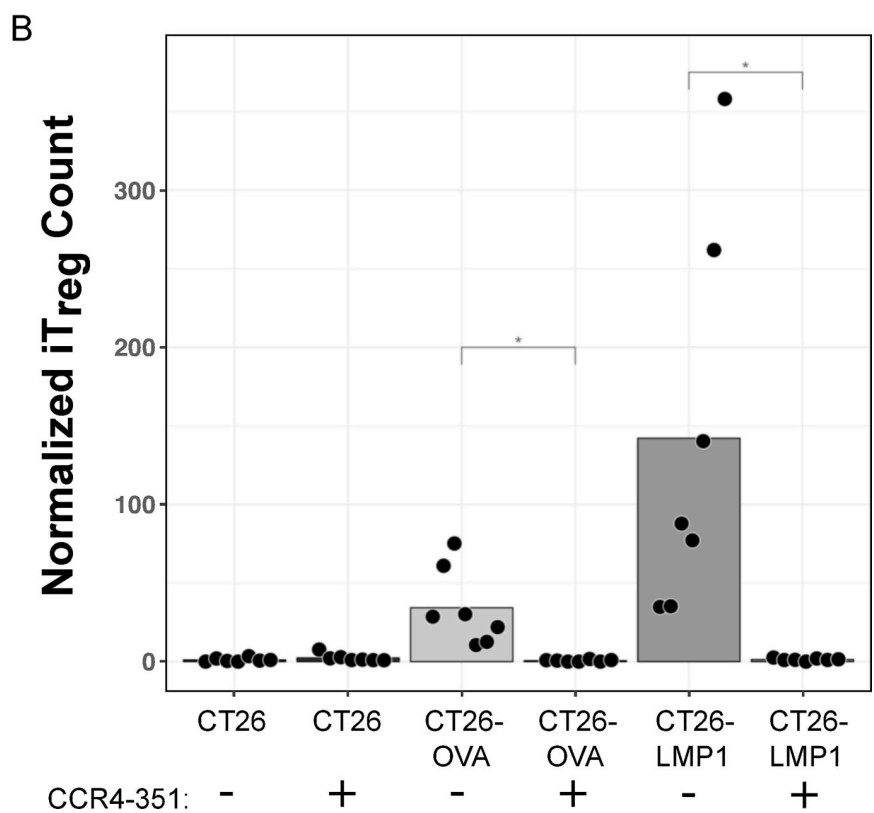

**Fig 6.** *In vivo* **CCL22 expression and T$_{reg}$ infiltration is associated with LMP1 in a mouse epithelial tumor model.** Murine colon CT26 cells, along with CT26 engineered to express OVA (CT26-OVA) or LMP1 (CT26-LMP1) were inoculated into mice to produce syngeneic tumors. (A) CCL22 concentrations were measured by ELISA in the *in vitro* cell lines and harvested tumors (n = 10). (B) GFP$^+$ iT$_{reg}$ were transferred into tumor-bearing mice, treated with vehicle (-) or CCR4-351 (+), and quantified in harvested tumors by FACS after 7 days (n = 7).

matched IO approaches. Support for this conjecture comes from multiple observations of high levels of infiltrating T$_{reg}$ in different types of EBV$^+$ tumors (this work and [14,16,17,29,40]). T$_{reg}$ are a type of CD4$^+$ lymphocyte that tempers inflammation by suppressing the activity of cytolytic CD8$^+$ T cells. A pan-tumor analysis shows that T$_{reg}$ levels track those of CD8$^+$ cells, suggesting an adaptive immune resistance mechanism that acts as a negative feedback process in the TME. Thus, reducing the activity or number of T$_{reg}$ may help to drive antitumor immune responses in these tumor types.

The chemokines CCL17 and CCL22 are potent activators of the chemokine receptor CCR4, and this CCL17/22/CCR4 axis may be the major mechanism for recruiting T$_{reg}$ into tumors (this work and [41–43]). Although TGFβ can support the conversion of CD4$^+$ T cells to T$_{reg}$ as well as their subsequent proliferation, existing data suggests that migration rather than conversion/expansion is the key driver of T$_{reg}$ numbers in tumors [23,44]. Here, we show that out of a variety of human EBV$^+$ tumor-derived cells lines, only those of B-lymphoma origin that express LMP1 also secreted both CCL17 and CCL22 at high levels *in vitro*. This is in agreement with other work such as in age-related EBV$^+$ B-cell lymphoproliferative disorder (ALPD) [19]. Reducing LMP1 mRNA levels in these cells via siRNA reduced CCL17/22 expression. LMP2A, which has been shown to cooperate with LMP1 for its activity [24,25], also affected CCL17/22 production but did not appear to be sufficient in these cells. Since LMP1 is believed to mimic CD40 activation in B cells [27,28], a process which normally leads to CCL17/22 expression, there is a clear mechanism driving this pathway.

Less clear has been the mechanism for CCL17/22 expression in EBV$^+$ tumors of epithelial origin. The EBV$^+$ gastric carcinoma cell line we tested, NCI-N87, did not produce either chemokine *in vitro* and RNA expression data for the NPC cell line C666-1 showed barely detectable chemokine levels [45]. However, we observed that the most common EBV$^+$ epithelial tumor types, EBV$^+$ gastric carcinoma and nasopharyngeal carcinoma, are among the highest expressors of both chemokines. Strikingly, these appear to also be the highest T$_{reg}$-infiltrated tumors based on levels of FOXP3 expression across nearly 10,000 disparate samples. Matching these tumor expression data, we observed chemokine expression by RNA ISH in NPCs and in NPC xenografts. Unlike the expression pattern we observed in Hodgkin lymphomas, where CCL17/22 expression was strongly linked to EBV-positivity, the NPCs showed chemokine expression that appeared to be a combination of tumor cell-intrinsic and expression by tumor-infiltrating EBV$^-$ cells. A mixed human/mouse xenograft system allowed us to more cleanly observe the sources of chemokine expression. In fact, both human and mouse CCL22 expression was observed, and this expression was localized, respectively, to tumor cells and regions of host cell infiltration.

LMP1 is not expressed in all NPCs [46,47] and is detected in only one of the NPC xenografts we tested (C15; [32]). Interestingly, LMP1-negative NPCs have been shown to have genetic alterations that can mimic the effects of LMP1 expression such as the genetic TRAF3 inactivation in C666-1 cells that mimics TRAF3 sequestration by LMP1 [48–50]. The better-established NFκB-activating role of LMP1 in EBV$^+$ lymphomas led us to test the effect of engineered LMP1 expression on CCL17/22 expression in an exemplar epithelial tumor. We engineered the mouse colon cancer cell line CT26 to express LMP1 and assayed its chemokine production. Although no CCL17/22 expression was observed *in vitro* by either the parental

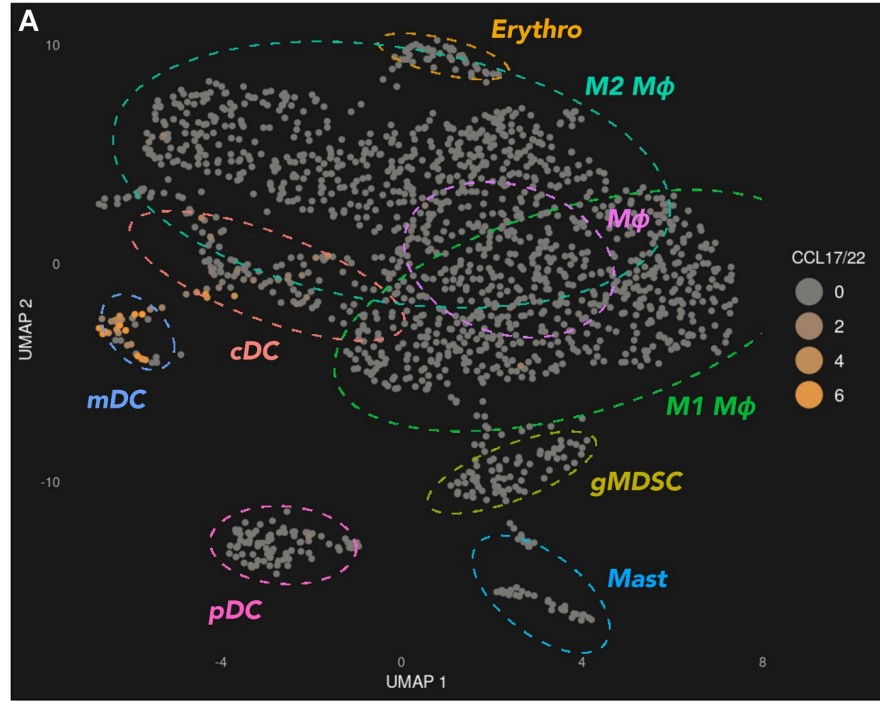

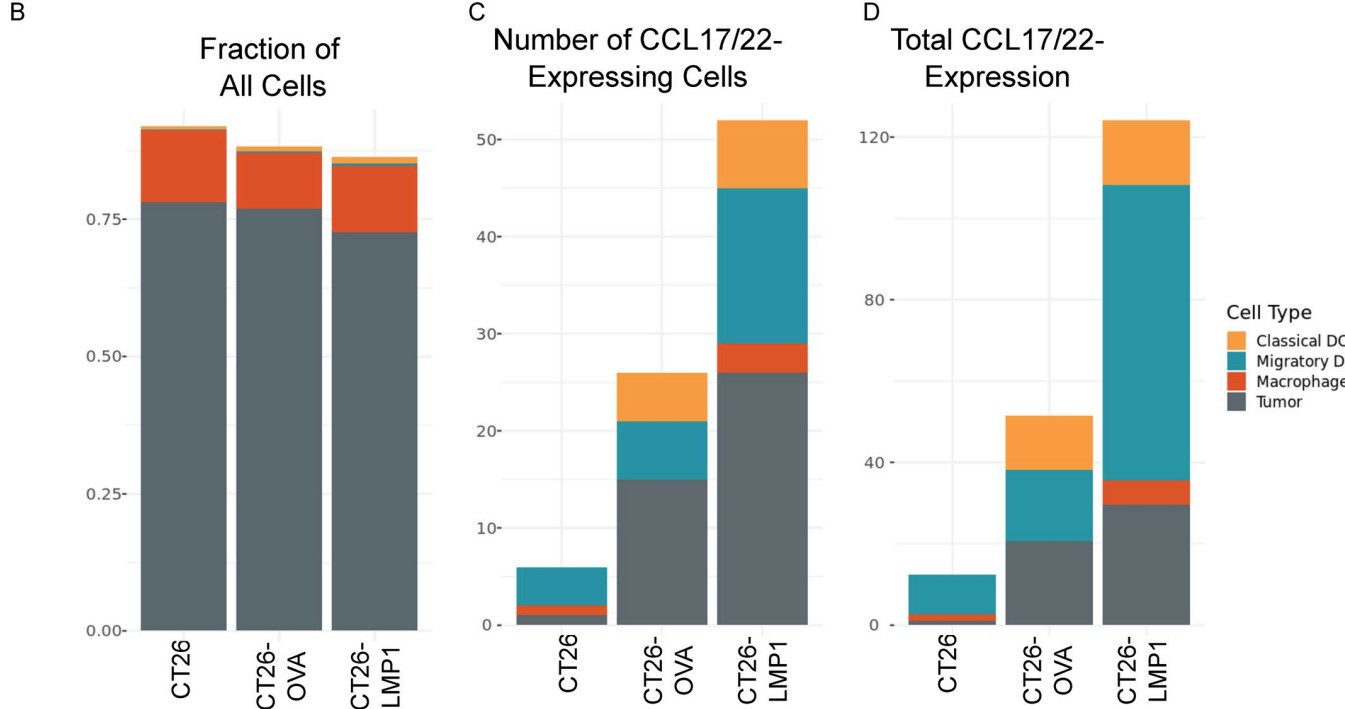

**Fig 7. CCL17/22 expression is restricted to tumor and myeloid cells and increases with LMP1 expression.** (A) Combined CCL17/22 expression is indicated on a 2D UMAP projection of all myeloid cells from CT26, CT26-OVA, and CT26-LMP1 tumors as identified by single-cell RNA Seq. CCL17/22 expression is in Transcripts per Million (TPM). Ellipses indicate cell types: Erythro = Erythrocyte; Mast = Mast Cell; gMDSC = granulocytic Myeloid-Derived Suppressor Cell; mDC = migratory Dendritic Cell; cDC = classical Dendritic Cell; pDC = plasmacytoid Dendritic Cell; M1 Mϕ = M1 Macrophage; M2 Mϕ = M2 Macrophage; Mϕ = Macrophage, unknown subtype. Stacked barcharts show, for each tumor type: (B) the fractions of all cells from the five CCL17/22-expressing cell types; (C) the total number of cells from B with CCL17/22 TPM > 1.0; (D) the sum of CCL17/22 TPM from the cells in (C).

CT26 cell line or by CT26-LMP1, strong chemokine expression was seen *in vivo* with CT26-LMP1 when the cell lines were grown as syngeneic tumors. Examining the tumors by single cell RNA-Seq revealed expression from the CT26-LMP1 tumor cells themselves as well as a major contribution from dendritic cells, especially from migratory DCs. These migratory DCs are a class of myeloid cells that bring antigen from peripheral tissues to lymph nodes [51], and their increased expression of the suppression-related chemokines CCL17 and CCL22 may play an important role in changing immune responses to EBV-infected tumors. DC migration is complex and only partially understood, with a variety of chemokines, including CCL19 and CCL21, and small molecules such as leukotriene B4 and oxysterols (which are bound by the intriguingly-named Epstein Barr Virus Induced Gene 2 receptor) acting as possible DC attractants [52,53]. Although we did not observe significant changes in the best known chemokine DC attractants, their general low levels of expression and restriction to specific cell compartments may have led to a lack of sensitivity in detection of such expression changes.

Edwards *et al.* grew AGS, a human EBV- gastric cancer cell line, *in vitro* and *in vivo* in mice, along with an LMP1-expressing engineered version of AGS and an EBV-infected version of AGS as well as the human NPC xenografts C666, C15, and C17, and compared these cell lines and conditions by transcriptional and protein profiling[54]. While these experiments comparing the properties of EBV-related cell lines *in vivo* and *in vitro* are broadly similar to the experiments described herein, Edwards *et al.* did not identify the same biological processes we did. While one would hope that an upregulation of CCL17/22 would have been corroborated in that study, it is important to consider the major differences between these two studies: Edwards *et al.* looked at human cells in immune-deficient mice while we analyzed a mouse cell line in immune-competent mice; Edwards *et al.* used bulk, rather than single cell, RNA Sequencing that would likely miss changes due to rarer cells such as chemokine-expressing DCs; the differential gene expression analysis reported in Edwards *et al.* focused on the human gene expression changes as opposed to changes in the tumor-infiltrating mouse cells. Edwards *et al.* serves as an informative study of the changes to tumor-intrinsic pathways triggered by EBV, such as *in vitro* regulation of gene transcription by miRNAs[54].

Although the experiments reported here and mechanistic studies [19,20,50] show how EBV-derived LMP1 can lead to increased chemokine expression, the mechanism for the tumor-extrinsic increase in CCL17/22, shown here in a variety of settings, is less clear. We have shown that LMP1 expression leads to both an increase in the number of chemokine-expressing DCs and the level of chemokine expression in these cells. In contrast, no significant increase in mDC or cDC number was observed in the antigenicity-control model, CT26-OVA, and a more modest increase in chemokine expression was observed. Whether the difference between CT26-OVA and CT26-LMP1 is one of quantity, with perhaps LMP1 being more antigenic than OVA, or quality, where LMP1 participates in a specific biological pathway, is unknown. Regardless of which or both of these mechanisms are in play, we postulate that the net effect of EBV-LMP1 is a marked increase in CCL17/22 production that fosters an immune-suppressive environment beneficial to the EBV+ tumor. Further, regardless of mechanism, antagonism of CCR4 by CCR4-351, a novel, oral specific small-molecule inhibitor, completely blocked the new infiltration of $T_{reg}$-polarized cells in our mouse model.

In summary, multiple lines of evidence suggest that suppression of productive inflammation by $T_{reg}$ may be a common mechanism employed by EBV+ tumors of both lymphocytic and epithelial origins. In both cases, tumor-produced chemokines CCL17 and CCL22 trigger $T_{reg}$ migration by activating the CCR4 chemokine receptor, and, in both cases, the viral protein LMP1 may be central to this process. In lymphomas, LMP1 has been shown to coopt existing B cell-intrinsic pathways to directly upregulate the chemokines. In contrast, chemokine production in epithelially-derived EBV+ tumors is likely due to a combination of both tumor-

intrinsic and tumor-extrinsic mechanisms. LMP1 and/or other viral proteins may lead to the indirect production of CCL17/22 in EBV$^+$ tumors via recruitment of infiltrating cells such as dendritic cells followed by tumor-intrinsic CCL17/22 expression in response to the activity of these infiltrating immune cells. These data suggest that blocking the T$_{reg}$ CCR4 receptor, such as with the selective antagonist CCR4-351, may be an effective way to potentiate antitumor inflammation and be an important part of a pan-EBV$^+$ tumor therapy.

## Material and methods

### Ethics statement

Propagation in nude mice was done with the approval of the Gustave Roussy Ethics Committee for Animal Experimentation (APAFIS#1605-2015090216498538v2 –November 26, 2015).

### Animal studies

Six- to eight-week old female mice were obtained from JAX Mice and Services (Bar Harbor, ME): Balb/cJ (000651), C57BL/6-Tg(Foxp3-GFP)90Pkraj/J (023800), Foxp3-GFP-Balb/cJ (006769) and NOD.CB17-Prkdcscid/J (001303). Animals were randomized between groups and none were excluded after randomization. Experiments were conducted in compliance with internal protocols reviewed and approved by the IACUC at RAPT Therapeutics.

### CCR4 antagonist

CCR4-351 was designed, synthesized and characterized at RAPT Therapeutics (Compound 38 in Robles et al. [55]).

### Cells and culture conditions

Raji, Daudi, NC-37, NCI-N87, Jijoye, Ramos, CT26, CT26-OVA, and CT26-LMP1 cells were grown in complete RPMI-1640 (Basal medium with 1% NEAE, 1% Penicillin-Streptomycin, 100IU/mL L-glutamine), 10% FBS and plated at 0.5x10$^6$/mL in a T-25 flask. P3HR1 and KATO III were grown as above, but with 20% FBS. Namalwa were cultured in RPMI-1640 with 2 mM L-glutamine adjusted to 1.5 g/L sodium bicarbonate, 4.5 g/L glucose, 10 mM HEPES, and 1.0 mM sodium pyruvate, 92.5%; fetal bovine serum, 7.5%. Media was changed every 24hr for treatments longer than 72hr. CCRF-CEM cells were seeded at 0.2–0.3 million cells/mL and cultured in complete RPMI-1640, 10% FBS. Cells were obtained from the American Type Culture Collection (ATCC), frozen at passage 3–5, and used at passage 4–6. CT26-OVA and CT26-LMP1 were generated by stable transduction with adenovirus carrying chicken ovalbumin (OVA) or EBV LMP1 under control of the CMV promoter. Supernatants were collected 24hr after the final split.

### Western blotting

Whole-cell lysates in RIPA (with phosphatase and protease inhibitors) were boiled for 10 minutes in LDS buffer with reducing agent, separated by 4–12% SDS-polyacrylamide gel electrophoresis, and transferred to nitrocellulose membrane. Membranes were blocked with 5% milk in Tris-buffered saline-Tween 20 and incubated overnight at 4°C with primary antibodies: anti-LMP1 (A301-957A, ThermoFisher, TFS), anti-LMP2A (MA1-81921, TFS), anti-EBNA1 (sc-81581, Santa Cruz Biotechnology), anti-HSP90 (4874S, New England Biolabs, NEB), or anti-Actin (4970S, NEB). Membranes were washed and incubated with HRP-conjugated secondary antibodies for 1hr at room temperature (RT), followed by rewashing and visualization

with enhanced chemiluminescence reagent. Densitometry was done with Li-Cor Image Studio Lite software.

### ELISA

Chemokines were assayed using ELISA kits according to manufacturer protocols for CCL17 (human DDN00, mouse DY529-05), CCL20 (human DM3A00), CCL22 (human DMD00, mouse MCC220); all from R&D Systems.

### siRNA Transfection

Raji cells were seeded at $10^6$ cells/well in 6-well plates to reach 80% confluence the following day. To transfect, 10 μL lipofectamine 2000 (Invitrogen) in 250 μL Opti-MEM and 40 μM of siRNA in 250 μL Opti-MEM were incubated separately for 4 minutes at RT, combined, mixed gently, incubated for 20 minutes at RT, and added to cells in 2 mL RPMI 1640 for 4hrs before washing. siRNA sequences were: LMP1 GGAAUUUGCACGGACAGGCUUUU; LMP2A AACUCCCAAUAUCCAUCUGCUUU; LMP1 control AACUCCCAAUAUCCAUCUG CUUU; LMP2A control CUCCCAAUUAGCAUCUGCUTTUU (nucleotides 9 and 11 switched in negative control siRNAs [56]).

### Human T$_{reg}$ *in vitro* generation

Human T$_{reg}$-polarized CD4$^+$ cells (induced T$_{reg}$; iT$_{reg}$) were generated as previously described [42]. Routinely, >90% of CD4$^+$ cells expressed Ccr4 and 30–60% expressed FoxP3. iT$_{reg}$ suppressed CD8$^+$ T cell activation at levels comparable to natural T$_{reg}$ isolated from human PBMCs.

### Mouse GFP T$_{reg}$ *in vitro* generation

Single-cell suspensions were prepared from spleen and lymph nodes from 6-8-week-old C57BL/6-Tg (Foxp3-GFP) 90Pkraj/J mice (*in vitro* studies) or 6–8-week-old Foxp3-GFP-Balb/cJ (*in vivo* studies). Red blood cells were removed using 1x ACK lysis buffer (Gibco, A1049201). CD4 cells were isolated by depleting CD25$^+$ cells using the CD25 MicroBead Kit (130-091-072, Miltenyi Biotec) and enriched using a CD4 T Cell Isolation Kit (130-104-454, Miltenyi Biotech). Cells were cultured in anti-mouse-CD3-coated plates in complete DMEM (MT10013CV, TFS) with 1% NEAE, 1% Penicillin-Streptomycin, 100IU/mL, L-glutamine, 10% FBS. Complete media was supplemented with β-Mercaptoethanol (21985023, TFS), 5 ng/mL of TGF-β (7666-MB-005, R&D Systems), IL-2 (402-ML-020, R&D Systems), anti-IFN-γ (BE0055, Invivogen), anti-IL-4 (BE0045, Invivogen), and 1 μg/mL anti-CD28 (16-0281-86, TFS); and cultured in 5 μg/mL anti-CD3 (16-0031-85, TFS) coated plates at a concentration of $10^6$ cells/mL. On day 3, cells were cultured in RPMI medium with 20 ng/mL IL-2. Cells were harvested on day 7 for studies. Over 90% of T$_{reg}$-polarized cells expressed CD25 and GFP.

### Raji/Daudi tumor models

NOD-SCID (6–8 months) mice were injected subcutaneously with $2x10^6$ cells in 100 μL PBS with 100 μL Matrigel (CB40234A, TFS). Tumors were harvested at 200–400 mm$^3$ and lysed in 25 mM HEPES, 150 mM NaCl, 1% Triton X-100, 5 mM EDTA at pH 7.4 with metallic beads with pulsed shaking. Lysates were quantified via Bradford assay and normalized to tumor weight.

### Human $T_{reg}$ migration study–Raji/Daudi

When Raji tumors reached volumes of 200–400 mm$^3$, mice were treated with 50 mg/kg CCR4-351 and 3 hours later injected with 8x10$^6$ human iT$_{reg}$ (>90% purity by Flow analysis of CD4, CD25 and CCR4). Tumors were harvested after 7 days into digestion buffer with DNase (89836, TFS) and lysed in gentleMACS C tubes (130-093-237, Miltenyi Biotec) using the Gentle MACS Octodissociator. Staining was performed with: TruStain FcX (anti-mCD16/32) antibody (101320, Biolegend), human TruStain FcX (422302, Biolegend), hCD4 APC Cy7 (317450, Biolegend), hCD45 PE Cy7 (368532, Biolegend), mCD45 BV510 (103138, Biolegend), hCD19 APC (392504, Biolegend), hCCR4 APC (59407, Biolegend), 7-AAD live/dead stain PERCP Cy5.5 (420404, Biolegend). Data was collected on BD Fortessa and analyzed using FlowJo software.

### *In vitro* chemotaxis

Assays were performed using the ChemoTX migration system with a 5 µm pore size PCTE membrane (106–5, Neuro Probe). CCRF-CEM cells were resuspended at 2x10$^6$ cells/mL in human serum. CCR4-351 (300 nM) or DMSO were added to a DMSO concentration of 0.25% (v/v) followed by a 30-minute preincubation. 29 µL of recombinant hCCL22 (diluted to 0.9 nM in 1xHBSS with 0.1% BSA) or supernatant from cultured cells was dispensed in the lower wells. PCTE membrane was placed onto the plates and 50 µL of the CCRF-CEM cell/compound mixture was transferred on top. Plates were incubated at 37˚C, 100% humidity, 5% CO$_2$ for 60 minutes, then the membranes were removed and 15 µL Cell Titer Glo was added to lower wells. Luminescence was measured using an Envision plate reader (PerkinElmer).

### Mouse *in vivo* $T_{reg}$ migration

After CT26, CT26-LMP1, and CT26-OVA tumors reached 200–300 mm$^3$, mice were given 50 mg/kg CCR4-351 or vehicle orally. Three hours later, mice were injected intravenously with GFP$^+$ iT$_{reg}$ at 97% purity and 27% CCR4-positivity. Tumors were harvested after 7 days, during which CCR4-351 was dosed orally daily, and incubated in digestion buffer with DNAse and lysed in Miltenyi C tubes using the Gentle MACS Octodissociator. A single cell suspension was prepared from spleens using syringes, filtered and stained for: TruStain FcX anti-mouse CD16/32 antibody (101320), CD4 APC Cy7 (100414), mouse CD45 BV510 (103138), mouse CD8 PE Cy7 (100722), GFP-FoxP3-FTIC, 7-AAD Live dead stain PERCP Cy5.5 (420404), mouse CCR4 APC (359410), CD11c BV605 (117334), MHC II (I-A/I-E Antibody) APC (107614), CD11b PE (101208), F4/80 (123141), and M1/70 BV785 (101243); all from BioLegend. Data was collected on BD Fortessa and analyzed using FlowJo.

### Bulk RNA-Seq

Solid tumor TCGA and TARGET RNA-Seq datasets were downloaded from the UCSC Xena data hub [57] on June 18$^{th}$, 2017. NPC datasets GSE102349 [58] and GSE68799 were downloaded from NCBI GEO and processed with Kallisto [59]. Counts across all data sets were quantile-normalized using preprocessCore [60]. EBV status for GC was obtained from cBioPortal [61]. S1 Table shows tumor-type abbreviations.

### Single cell RNA-Seq

After tumors reached 200–300 mm$^3$, they were collected and digested with collagenase buffer in a 37˚C bath, pipetting every 10 minutes to dissociate. Cells from 5 tumors were pooled per sample. Cell surface protein feature barcoding (CITE seq) antibody (BioLegend TotalSeq-A)

incubation was performed per manufacturer protocols, and cells were processed for 10X Chromium 3' RNA seq (v3) chemistry reagents with slight modifications for CITE seq. FASTQ files were aligned to the mouse mm10 genome with CellRanger (v3.1.0). Data was analyzed in R [62] using Seurat (v3.2.2) [63] and tidyseurat [64] packages. scRNA data deposited as PRJNA736082 to SRA.

## RNA *in situ* hybridization

HL biopsies were purchased from US Biomax (HL801a, Rockville, MD) as a tumor microarray. NPC tumor slices were purchased from ACD Bio (AB, Newark, CA). FFPE-preserved paraffin-embedded samples were probed on the RNAscope platform [31]. Two EBER1 double-Z probe pairs (310271-C2, AB), 20 CCL22 probe pairs (468701, AB), 13 CCL17 probe pairs (468531, AB), and 20 FOXP3 probe pairs (418471-C2, AB) were used. Following hybridization and chromogenic detection, hematoxylin counterstaining was followed by blueing. Imaging was done on the Leica AT2 scanner and analyzed with Indica Labs (Albuquerque, NM) HALO software.

## NPC Xenografts

C15, C17, and C18 are patient-derived xenografts (PDX) from cells propagated solely in nude mice [65,66]. The C666-1 NPC tumor line was first established as a PDX and later as a cell line propagated *in vitro*[67] and recently re-implanted in nude mice by one of us (PB).

## Statistics

Significance tests used for data in plots are two sample Student's t-tests unless otherwise noted. Significance for LMP proteins and chemokines reported for Fig 1A was calculated by the R "stats::cor.test" function using Pearson correlations and two-sided hypothesis testing. A repeated measures ANOVA was used for data in Fig 2D. Significance codes for p-values are: **** = $p < 0.0001$, *** = $p < 0.001$, ** = $p < 0.01$, * = $p < 0.05$.

## Supporting information

**S1 Fig. Quantitation of LMP1, LMP2A, and CCL22 expression by various EBV cell lines.** (A) Western blots on 50 μg of protein lysate from 9 human cell lines were probed for LMP1, LMP2A, and HSP90 as indicated. Ramos and Namalwa were run on a separate gel. (B) RT-PCR for LMP1/2 in three cell lines confirms detection of LMP transcript in NCI-N87. (C) Supernatants from increasing numbers of seeded Raji cells (50 to 2000 thousand cells per well) or from 2000 Daudi cells grown for 24 hours, or a standard of 2000 pg/mL CCL22 quantitated by CCL22 ELISA. Where bands from the same Western blot have been reordered to match across analytes, white gaps are shown–LMP1 and HSP90 were probed on a separate blot from LMP2A. Ramos and Namalwa lysates were run on a separate gel, visually separated by a black border, from the other lysates.
(TIF)

**S2 Fig. No difference in tumor growth was observed between Raji and Daudi xenograft tumors.** Raji and Daudi Xenografts were established in NOD/SCID mice to measure chemokine production and iT$_{reg}$ migration (Fig 2C and 2D). Tumor size was measured by calipers regularly over 30 days post- inoculation with $2 \times 10^6$ of Raji or Daudi cells, as indicated. Curves show the means and standard deviations for calculated tumor volumes from 5 mice per tumor type.
(TIF)

**S3 Fig. Controls support elevated FOXP3 and CCR4 ligand expression in EBV⁺ tumor types.** Data is shown as in Fig 3, with the addition of control "housekeeping" genes β-Actin (ACTB) and TATA-Box Binding Protein (TBP). Tumor types are sorted by increasing median expression and plotted as $\log_2$ Transcripts per Million for each gene. Tumor abbreviations are defined in S1 Table.
(TIF)

**S4 Fig. Dimensionality reduction visualization shows that EBV⁺ tumors types are well integrated with other tumors.** Principal Component Analysis was performed on the tumor expression data from which Fig 3A and S3 Fig are derived, with the first two principal components (PC.1 and PC.2) plotted along the X and Y axes, respectively. EBV⁺ tumor types are shown in filled circles, all others in open circles. The NPC and GC_EBV samples are distributed amongst other tumor types, and in close proximity to EBV⁻ Gastric (GC) and Head & Neck (HNSC) carcinomas. Tumor abbreviations are defined in S1 Table.
(TIF)

**S5 Fig. RNA *in situ* hybridization on additional Hodgkin Lymphoma (HL) samples.** Top left: 20x magnification of staining for EBER1 (red) and CCL22 (cyan); top right: 20x EBER1 (red) and CCL17 (cyan); bottom left and right: 40x magnification of the upper images. Nuclear haematoxylin staining is in blue. Arrows highlight CCL17 and CCL22 in the fourth biopsy sample. CCL17 and CCL22 signals can be seen in all but the 5th biopsy sample.
(TIF)

**S6 Fig. RNA *in situ* hybridization shows tumor-intrinsic and -extrinsic expression of CCL22 in Nasopharyngeal carcinoma xenografts.** Representative images of duplex RNA *in situ* hybridization of a (A) C17 NPC xenograft and a (B) C18 NPC xenograft are shown. Images shown are of corresponding serial sections: H&E (top right); EBER1 (cyan) and human CCL22 (red) (top and bottom left); and mouse CCL22 (cyan) and human CCL22 (red) (bottom right). Arrows highlight select human CCL22 (bottom left panels) and mouse CCL22 (bottom right panels) staining. Staining in bottom panels is digitally enhanced (unenhanced versions in S8A and S8B Fig). Nuclear haematoxylin staining appears in pale blue.
(TIF)

**S7 Fig. Reference unprocessed images of Nasopharyngeal carcinoma RNA *in situ* hybridizations.** Representative images of RNA *in situ* hybridization (ISH) of (A) Hodgkin lymphoma (HL) and (B) nasopharyngeal carcinoma (NPC) samples probed for EBER1 (red) and CCL22 (cyan) are shown. (C) A matched section serial to that in (B) was probed for FOXP3 (red) and CCL22 (cyan). Yellow boxes in lower magnification views (left) indicate sources of magnified regions shown on right. Nuclear haematoxylin staining is shown in pale blue. These unprocessed images exported by HALO software correspond to the color-enhanced versions shown in Fig 4.
(TIFF)

**S8 Fig. Reference unprocessed images of Nasopharyngeal carcinoma xenograft RNA *in situ* hybridizations.** Representative images of RNA *in situ* hybridization (ISH) of a (A) C15 NPC xenograft and a (B) C666-1 NPC xenograft are shown. Images shown are of corresponding serial sections: H&E (top right); EBER1 (cyan) and human CCL22 (red) (top and bottom left at two magnifications; and mouse CCL22 (cyan) and human CCL22 (red) (bottom right). Nuclear haematoxylin staining shown in pale blue. These unprocessed images exported by HALO software correspond to the color-enhanced versions shown in Fig 5.
(TIFF)

**S9 Fig. iT$_{reg}$ migration to spleen is unaffected by treatment or tumor type.** The numbers of iT$_{reg}$ from spleen, identified as CD4$^+$ GFP$^+$ cells, normalized to total CD45$^+$ cell count, were quantified by FACS analysis.
(TIF)

**S10 Fig. UMAP projections of myeloid cells from single cell RNA Sequencing.** The UMAP 2D projection of all myeloid-compartment cells is colored by combined CCL17 and CCL22 expression (TPM) and faceted by tumor (A), colored by cell cluster (B), or colored by tumor source (C). CT = CT26, OV = CT26-OVA, LM = CT26-LMP1
(TIF)

**S11 Fig. UMAP projections of lymphoid cells from single cell RNA Sequencing.** The UMAP 2D projection of all lymphoid-compartment cells is colored by cell cluster. The lymphoid UMAP projection is colored by cell cluster (A) or by tumor source (B). CT = CT26, OV = CT26-OVA, LM = CT26-LMP1
(TIF)

**S12 Fig. UMAP projections of tumor cells from single cell RNA Sequencing.** The UMAP 2D projection of all myeloid-compartment cells is colored by combined CCL17 and CCL22 expression (TPM) (A), colored by cell cluster (B), or colored by tumor source (C). CT = CT26, OV = CT26-OVA, LM = CT26-LMP1
(TIF)

**S13 Fig. Published gene signatures applied to Dendritic Cell (DC) subsets support labeling of classical and migratory DC.** Gene signatures from Binneweis *et al*, Maier *et al*, and Miller *et al* (S4 Table) were applied to the three DC-like clusters identified in this study. A robust Z-transform was used on the relevant genes across all myeloid cells, then the average Z scores for the genes in each signature were plotted for myeloid cluster 1 (migratory DC, mDC), myeloid cluster 7 (classical resident DC, cDC), and myeloid cluster 8 (plasmacytoid DC, pDC). The cDC cluster stood out for Binnewies signature 3 (resident CD11b$^+$ cDC2), Binnewies signature 7 (resident CD8a$^+$ cDC1), and the Miller cDC signature. The mDC cluster stood out for Binnewies signature 1 (migratory CD103$^+$ cDC1), Binnewies signature 2 (Langerhans cell), Miller mDC, and the Maier mregDC signature. The pDC assignment is supported by expression of Tlr7 and Tlr9 in this cell cluster.
(TIF)

**S14 Fig. Combined CCL17 and CCL22 expression per cell for all CCL17/22-expressing cell types across all tumor types.** Combined CCL17 and CCL22 expression for each CCL17/22-expressing cell from each tumor sample are plotted as Transcripts per Million (TPM). Short horizontal lines indicate median expression values per population. A TPM of 1.0 (dashed line) was arbitrarily considered to be "productive" expression for further filtering. CT = CT26, OV = CT26-OVA, LM = CT26-LMP1
(TIF)

**S1 Table. Tumor Code Lookup Table.** A mapping of tumor type short abbreviates to full names.
(XLSX)

**S2 Table. Top 10 Cluster Markers.** Gene markers for each cell cluster in the single cell RNA-Sequencing data, as identified by the "FindAllMarkers" function of the Seurat analysis package. Additional details are given at the top of the table.
(XLSX)

**S3 Table. Significant Differential Tumor-Intrinsic Markers.** Gene markers distinguishing the tumor cells between tumor types in the single cell RNA-Sequencing data, as identified by the "FindAllMarkers" function of the Seurat analysis package. Additional details are given at the top of the table.
(XLSX)

**S4 Table. Dendritic Cell Signatures.** Gene signatures derived from the literature for different subtypes of Dendritic Cells (DCs). These are used to subclassify the DCs identified in the single cell RNA-Sequencing data.
(XLSX)

## Acknowledgments

This research was supported by RAPT Therapeutics, formally known as FLX Bio. Urvi Kolhatkar helped with densitometry. We thank Abood Okal for providing insight and expertise that greatly assisted the research. We are grateful to Steve Wong, Martin Brovarney, Justy Guagua, Jerick Sanchez, David Chan and Angela Wadsworth for providing their technical expertise during the course of this research. We are very grateful to Heather Milestone for chemokine-induction experiments that did not appear in the manuscript. Finally, we thank Brian Wong, CEO of RAPT Therapeutics, for manuscript review and continual support for this work.

## Author Contributions

**Conceptualization:** Aparna Jorapur, Pierre Busson, Dirk G. Brockstedt, Paul D. Kassner, Gene Cutler.

**Investigation:** Aparna Jorapur, Lisa A. Marshall, Scott Jacobson, Mengshu Xu, Sachie Marubayashi, Mikhail Zibinsky, Dennis X. Hu, Omar Robles, Jeffrey J. Jackson, Gene Cutler.

**Methodology:** Aparna Jorapur, Lisa A. Marshall, Mengshu Xu, Sachie Marubayashi.

**Resources:** Valentin Baloche, Pierre Busson.

**Supervision:** Pierre Busson, David Wustrow, Dirk G. Brockstedt, Oezcan Talay, Paul D. Kassner.

**Visualization:** Gene Cutler.

**Writing – original draft:** Gene Cutler.

**Writing – review & editing:** Pierre Busson, Dirk G. Brockstedt, Paul D. Kassner.

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
