## [Decision Letter · Decision Letter 0]

14 Jul 2021

Dear Dr Cutler,

Thank you very much for submitting your manuscript "EBV+ tumors exploit tumor cell-intrinsic and -extrinsic mechanisms to produce regulatory T cell-recruiting chemokines CCL17 and CCL22" for consideration at PLOS Pathogens. As with all papers reviewed by the journal, your manuscript was reviewed by members of the editorial board and by several independent reviewers. In light of the reviews (below this email), we would like to invite the resubmission of a significantly-revised version that takes into account the reviewers' comments.

We cannot make any decision about publication until we have seen the revised manuscript and your response to the reviewers' comments. Your revised manuscript is also likely to be sent to reviewers for further evaluation.

Sincerely,

Christian Munz

Associate Editor

PLOS Pathogens

Blossom Damania

Section Editor

PLOS Pathogens

Kasturi Haldar

Editor-in-Chief

PLOS Pathogens

orcid.org/0000-0001-5065-158X

Michael Malim

Editor-in-Chief

PLOS Pathogens

orcid.org/0000-0002-7699-2064

Reviewer's Responses to Questions

**Part I - Summary**

Reviewer #1: Here the authors show that LMP1 fosters CCL17 and CCL22 expression in EBV+ tumor cell lines supporting the hypothesis that CCL17 and CCL22 expression induced by LMP1 contributes to EBV+ HL and NPC tumor biology. The chemokines CCL17 and CCL22 are usually secreted by APCs to attract T cells. CCR4, the receptor for CCL17 and CCL22, is expressed on tumor-infiltrating FOXP3+CD4+ Tregs, which play an important role in suppressing anti-tumor immunity. The authors suggest that, by secreting the two chemokines, EBV/LMP1+ tumors attract Tregs which suppress anti-tumor immunity and thereby foster immune escape of EBV/LMP1+ tumors. Accordingly, EBV+ HL tumor biopsies show significant coincidence of EBER, CCL17 and FOXP3 staining. Also in NPC biopsies chemokine expression and tumor-infiltrating Tregs were detected, although no clear coincidence with the presence of EBV was observed. EBV+ NPC xenografts showed tumor-intrinsic and -extrinsic expression of CCL22, which was not dependent on the presence of LMP1. Thus, no clear link between LMP1 expression, chemokine expression and Treg infiltration exists in NPC, in contrast to HL. To clarify a potential contribution of LMP1 to epthelial tumor biology and the attraction of infiltrating Tregs, the authors make use of an elegant colon tumor xenograft model. CT26 cells were engineered to express LMP1 (or OVA as control) and transplanted into mice. GFP-marked mouse iTregs were co-transferred and iTreg migration into the tumors was analysed. Notably, LMP1 induced a marked increase in Treg infiltration. In the CT26-LMP1 model, LMP1 enhanced chemokine expression induced by IFN gamma. Most notably, Treg migration into these tumors was blocked by the CCR4 small molecule inhibitor CCR4-351. The authors conclude that the CCR4 pathway is a novel target for pharmacological inhibition to restore anti-tumor immunity of EBV+ tumors. This claim is based on the following data: the CCR4 inhibitor CCR4-351 interfered with migration of CCRF-CEM T-lymphoblasts and iTregs induced by Raji supernatants in culture, the inhibitor was able to block iTreg migration into Raji or Daudi xenograft tumors in NOD-SCID mice, and, finally, CCR4-351 inhibited Treg migration into tumors formed by CT26-LMP1 cells.

Taken together, this is a very interesting manuscript on the biology of EBV/LMP1+ tumors highlighting the relevance of tumor-intrinsic, and in the case of NPC also -extrinsic, CCL17 and CCL22 expression for Treg attraction. However, the observation that LMP1 augments CCL17 and CCL22 expression in tumor cells is not entirely new. This mechanism likely plays an important role in suppressing anti-tumor immunity in EBV+ tumors, although the latter has not been proven formally. The highlight of this manuscript is clearly the identification of the CCR4 pathway as novel potential target for pharmacological immune therapy of EBV+ tumors.

Reviewer #2: Jorapur and colleagues use multiple lines of evidence (in vitro, in vivo. public databases, cell lines, patient samples, siRNA and small molecule blockade) to examine the basis for inflammation in EBV+ epithelial (gastric and NPC and LMP1 gene inserted) and B cell (Hodgkin and atypical Burkitt) malignancies. They conclude that via intrinsic (B cell) and epithelial (intrinsic and cell extrinsic) mechanisms, these tumours produce CCL17 and CCL22, which result in induction and migration of TREGs and migration of DCs. The data indicates that LMP1, possibly influenced by IFNg, are responsible for this induction, via activation of CCR4. The precise mechanism by which LMP1 upregulates these chemokines, and how EBV might induce cell extrinsic effects, is not explored.

Interestingly, a novel CCR4 inhibitor reduced TREG migration, suggesting a potential therapeutic avenue to reduce immune evasion via the tumour microenvironment.

The study is well conducted and extremely (particularly the results) well written. The data is novel, and provides an important counter-point to the paper of Edwards et al. As an interesting footnote, the gastric ca tumour NCI-N87, believed to be EBV-negative, was found to be EBV-pos.

**Part II – Major Issues: Key Experiments Required for Acceptance**

Reviewer #1: 1. sFig1 and lane 115 to 117) In contrast to previous reports the gastric carcinoma line NCI-N87 is now found to be EBV+. Are NCI-N87 cells used in this work derived from a certified source? This cell line should be verified.

2. Fig 1C) The mechanism of chemokine induction by LMP1/LMP2A is not sufficiently adressed. LMP2A seems to considerably contribute to CCL17 expression in Raji. Jijoye and NC-37 express low LMP2A levels. Is LMP1 sufficient to induce chemokines in these cells or may other mechanisms contribute? The authors should express LMP1, LMP2A and a combination of both in EBV-negative cells such as Ramos or NPC cells and analyse CCL17 levels. Also LMP1 with mutations in CTAR1 and/or CTAR2 could then be easily tested.

Reviewer #2: 1. It is not clear to this reviewer if the authors are stating that TREG tumour infiltration was induced or due to migration. If this cannot be inferred, please clarify this in the discussion.

2. Please discuss the mechanisms by which EBV might influence tumour extrinsic chemokine production.

3. Figure 4 and 5 are difficult to understand and view. Images (particularly on the right hand) are blurred, and colours are indistinct from each other, and the legends do not explain what we are meant to be observing. This needs considerable attention and re-working.

**Part III – Minor Issues: Editorial and Data Presentation Modifications**

Reviewer #1: 1. Statistics: Which T-Test was used?

2. 1A) Technical or biological replicates? Where do the LMP1/LMP2A expression data come from? Statistics is required.

3. sFig1A) Are all panels shown for a specific antibody derived from the same blot? Are the protein levels comparable?

4. Fig 1B) Does control LMP2A siRNA affect LMP1 expression?

5. Fig 4) What is known about the status of LMP1/LMP2A expression in the HL and NPC biopsies?

6. Fig 8) Statistics is required.

Reviewer #2: Was CCL17 proportional to the density of Raji cells in culture (as per CCL22).

Legend to Fig 2: explain CCRE CEM cells as CD4 lymphoblasts.

PLOS authors have the option to publish the peer review history of their article (what does this mean?). If published, this will include your full peer review and any attached files.

Reviewer #1: No

Reviewer #2: No
---

## [Decision Letter · Decision Letter 1]

1 Oct 2021

Dear Dr Cutler,

Thank you very much for submitting your manuscript "EBV+ tumors exploit tumor cell-intrinsic and -extrinsic mechanisms to produce regulatory T cell-recruiting chemokines CCL17 and CCL22" for consideration at PLOS Pathogens. As with all papers reviewed by the journal, your manuscript was reviewed by members of the editorial board and by several independent reviewers. The reviewers appreciated the attention to an important topic. Based on the reviews, we are likely to accept this manuscript for publication, providing that you modify the manuscript according to the review recommendations.

Sincerely,

Christian Munz

Associate Editor

PLOS Pathogens

Blossom Damania

Section Editor

PLOS Pathogens

Kasturi Haldar

Editor-in-Chief

PLOS Pathogens

orcid.org/0000-0001-5065-158X

Michael Malim

Editor-in-Chief

PLOS Pathogens

orcid.org/0000-0002-7699-2064

Reviewer Comments (if any, and for reference):

Reviewer's Responses to Questions

**Part I - Summary**

Reviewer #1: The authors have significanty improved the manuscript. I am now supportive for publication of the manuscript in PLoS Pathogens after may last minor specific points regarding statistics have been addressed (see below).

My previous second major point regarding the mechansim of CCL17 induction by LMP1 (and LMP2A) has, unfortunately, not been addressed by the authors. Such data would have added substantially to the manuscript, especially since CCL17 and CCL22 induction by LMP1 has been reported previously. However, I follow the authors' argument that the focus of the present manuscript is rather the dissection of immune evasion by EBV+ tumors and, especially, its pharmacological inhibition. The latter is a very important finding with regard to the potential treatment of EBV+ tumors and warrants publication in PLoS Pathogens.

Reviewer #2: Following revision, the study is sufficiently strong to recommend acceptance.

**Part II – Major Issues: Key Experiments Required for Acceptance**

Reviewer #1: (No Response)

Reviewer #2: I reviewed the authors responses to reviewers comments, primarily focussing on my requested revisions.

These have all been sufficiently addressed.

**Part III – Minor Issues: Editorial and Data Presentation Modifications**

Reviewer #1: 1. Statistics of Figure 1A): Significance of the claimed major differences should be verified by T-Tests and p-values should be given.

2. Statistics of Figure 8: Standard deviations should be shown. Is there a significant difference between CT26, OVA and LMP1? From this reviewer's point of view, this could be tested by ANOVA comparing the LMP1 with the CT26, OVA curves, or by T-Tests at a distinct IFNg concentration. In case of doubt, a statistician should be consulted.

Reviewer #2: Nil.

PLOS authors have the option to publish the peer review history of their article (what does this mean?). If published, this will include your full peer review and any attached files.

Reviewer #1: No

Reviewer #2: No

Figure Files:

Data Requirements:

Reproducibility:

References:

---

## [Editor Report · Decision Letter 2]

13 Dec 2021

Dear Dr Cutler,

We are pleased to inform you that your manuscript 'EBV+ tumors exploit tumor cell-intrinsic and -extrinsic mechanisms to produce regulatory T cell-recruiting chemokines CCL17 and CCL22' has been provisionally accepted for publication in PLOS Pathogens.

Best regards,

Christian Munz

Associate Editor

PLOS Pathogens

Blossom Damania

Section Editor

PLOS Pathogens

Kasturi Haldar

Editor-in-Chief

PLOS Pathogens

orcid.org/0000-0001-5065-158X

Michael Malim

Editor-in-Chief

PLOS Pathogens

orcid.org/0000-0002-7699-2064
---

## [Editor Report · Acceptance letter]

6 Jan 2022

Dear Dr. Kassner,

We are delighted to inform you that your manuscript, "EBV<sup>+<sup> tumors exploit tumor cell-intrinsic and -extrinsic mechanisms to produce regulatory T cell-recruiting chemokines CCL17 and CCL22," has been formally accepted for publication in PLOS Pathogens.

Best regards,

Kasturi Haldar

Editor-in-Chief

PLOS Pathogens

orcid.org/0000-0001-5065-158X

Michael Malim

Editor-in-Chief

PLOS Pathogens

orcid.org/0000-0002-7699-2064